# Chemical and Biological Characterization of Green and Processed Coffee Beans from *Coffea arabica* Varieties

**DOI:** 10.3390/molecules28124685

**Published:** 2023-06-10

**Authors:** Javier Gallardo-Ignacio, Anislada Santibáñez, Octavio Oropeza-Mariano, Ricardo Salazar, Rosa Mariana Montiel-Ruiz, Sandra Cabrera-Hilerio, Manasés Gonzáles-Cortazar, Francisco Cruz-Sosa, Pilar Nicasio-Torres

**Affiliations:** 1Departamento de Biotecnología, Universidad Autónoma Metropolitana-Iztapalapa, Av. Ferrocarril de San Rafael Atlixco No. 186, Col. Leyes de Reforma 1ª Sección, Iztapalapa, Mexico City 09310, Mexico; gajarig-07mx@hotmail.com; 2Centro de Investigación Biomédica del Sur, Instituto Mexicano del Seguro Social (CIBIS-IMSS), Argentina No. 1 Col Centro, Xochitepec 62790, Mexico; anisszg@gmail.com (A.S.); montielrmariana@gmail.com (R.M.M.-R.); gmanases@hotmail.com (M.G.-C.); 3Cafeticultores Mephaa de la Montaña, Paraje Montero, Malinaltepec 41500, Mexico; ooropezam05@gmail.com; 4Consejo Nacional de Humanidades, Ciencia y Tecnología (CONAHCyT), CONACYT, Laboratorio de Bromatología y Tecnología de Alimentos Universidad Autónoma de Guerrero, Av. Lázaro Cárdenas S/N, Chilpancingo de los Bravo 39086, Mexico; rsalazarlo@conacyt.mx; 5Laboratorio de Bromatología, Facultad de Ciencias Químicas, Benemérita Universidad Autónoma de Puebla, Av. San Claudio S/N Ciudad Universitaria, Puebla 72000, Mexico; sandra.cabrera@correo.buap.mx

**Keywords:** antioxidant, *C. arabica* beans, chlorogenic acid, caffeine, cytotoxicity, melanoidins

## Abstract

Coffee is one of the most consumed beverages in the world; its production is based mainly on varieties of the *Coffea arabica* species. Mexico stands out for its specialty and organic coffee. In Guerrero, the production is done by small indigenous community cooperatives that market their product as raw material. Official Mexico Standards stipulate the requirements for its commercialization within the national territory. In this work, the physical, chemical, and biological characterizations of green, medium, and dark roasted beans from *C. arabica* varieties were carried out. Analysis by HPLC showed higher chlorogenic acid (55 mg/g) and caffeine (1.8 mg/g) contents in the green beans of the Bourbon and Oro Azteca varieties. The caffeine (3.88 mg/g) and melanoidin (97 and 29 mg/g) contents increased according to the level of roasting; a dissimilar effect was found in the chlorogenic acid content (14.5 mg/g). The adequate nutritional content and the sensory evaluation allowed the classification of dark-roasted coffee as premium coffee (84.25 points) and medium-roasted coffee as specialty coffee (86.25 points). The roasted coffees presented antioxidant activity without cytotoxic effects; the presence of CGA and caffeine supports the beneficial effects of drinking coffee. The results obtained will serve as a basis for making decisions on improvements to the coffees analyzed.

## 1. Introduction

Coffee commercialization is mainly based on *Coffea arabica*, which accounts for 70% of global production. The cultivation of *C. arabica* occurs mainly in Latin American countries such as Colombia, Honduras, Peru, and Mexico, with a production of 13.8, 5.6, 4.45, and 3.7 million bags of 60 kg, respectively [1]. Peru and Mexico are recognized for their organic and high-altitude coffee production. In Mexico, 3.33% of total coffee production is organic, and 28,000 t are principally exported to the European Union [2].

In Mexico, there are more than 600 cooperatives dedicated to coffee cultivation whose producers have obtained certifications such as USDA Organic, Fair Trade, Shade Grown, Rainforest Alliance, and Small Producer to achieve new markets and offer their products [3]. Mexican coffee is cultivated in four main regions: the Gulf of Mexico, the Soconusco, the north-central part of Chiapas, and the Pacific Ocean slope; the latter includes Colima, Guerrero, Jalisco, Nayarit, and Oaxaca [4]. Chiapas State produces 41% of Mexican coffee. Of the total coffee production, 70% is exported to the United States, European Union, Japan, Cuba, and Canada as green coffee (85%), soluble coffee (12%), and roasted coffee (3%); the rest (30%) is for national sale [5].

In Guerrero, *C. arabica* varieties are cultivated in Costa Grande, Costa Chica, and the Región de la Montaña. The coffee cultivated in the “Región de la Montaña” is characterized by growing under shade, also known as “benefit”, and it is cultivated traditionally by people from the Mixtec and Tlapaneco ethnic groups, grouped in cooperatives [6]. The coffee produced by these cooperatives is sold to intermediary companies such as Asociación Rural de Interés Colectivo de R.L. (ARIC), CAFECO Agroindustrial del Pacífico S.A. de C.V., and the Unión of Ejidos y Comunidades Luz de La Montaña, A.C. [7]. More than 60% of the production is marketed as green coffee to Nestlé S.A. Company; a small amount is exported to Europe by the Network of Sustainable Self-Managed Farmers S.C (RASA); and the remaining ≈30% is destined for coffee shops highlighting Starbucks as the main buyer, self-service, and convenience stores (Oxxo, 7 Eleven, among others), as well as for local consumption [8].

Some producers have organized themselves to give added value to their coffee by looking for methods that improve the quality of their products and marketing; one of them is the Cooperative Cafeticultores Mephaa de La Montaña. This cooperative currently produces small batches of commercial and specialty coffee from mixtures of *C. arabica* varieties, Typica, Bourbon, and Oro Azteca, which are marketed at regional and national levels. The cooperative sells green and roasted coffee to coffee shops and roasters in the country (70%) such as Buzz Café, Bombilla Errante, Sonata tostadores, and Comercializadora Golmex de México S.A de C.V.; the rest is locally sold as roasted and ground coffee.

In Mexico, there are Official Mexico Norms such as NMX-F-013-SCFI-2010 and PROY-NOM-255-SE-2021 that establish physical, chemical, and nutritional specifications to market roasted and ground coffee within the national market. The Cooperative Cafeticultores Mephaa de “La Montaña” coffee production is carried out on a small scale and organically. To expand its market, it is necessary to carry out studies that contribute to improving yields and cost reduction [9,10].

The main compounds related to the cup quality of the coffee drink, which gives it astringency and flavor, are chlorogenic acid (CGA), caffeine, caffeic acid, ferulic acid, vanillic acid, cinnamic acid, trigonelline, and volatile compounds such as furans, pyridines, pyrazines, and pyrroles [11,12].

The major compounds present in the coffee are CGA, caffeic, ferulic, catechin, epicatechin, and anthocyanin phenolic compounds; caffeine and trigonelline alkaloid compounds; and those that form during roasting, such as melanoidins and acrylamide (Figure 1) [11,12]. The Official Mexican Norms specify the caffeine and acrylamide levels required to market roasted and ground coffee. Furthermore, these compounds have important biological activities such as antioxidants, caffeine as a neurostimulator, CGA as an anti-inflammatory, and glucose and lipid metabolism [13,14,15].

The Cooperative Cafeticultores Mephaa de “La Montaña” has implemented a physical analysis of green beans as a first step in improving their commercial coffees. To address this point, in the present study, it was proposed to determine the nutritional and chemical specifications of green, medium, and dark roasted coffee beans that are currently marketed in addition to the antioxidant effect, as well as the cup quality of coffee drinks obtained from these commercial coffees.

## 2. Results and Discussion

The organic coffee of the State of Guerrero is known for its natural dry process, which preserves the pulp. In La Región de la Montaña, in the State of Guerrero, coffee is produced by people of the Mixtec and Tlapaneca ethnic groups, whose crops are mainly the Typica, Bourbon, Caturra, and Mundo Novo varieties of *C. arabica*, although hybrid varieties are being promoted such as Colombia, Costa Rica 95, Oro Azteca, and Sarchimor. The Cooperative Cafeticultores Mephaa de “La Montaña” cut the ripe cherries and processed them in an artisanal way; cherries are dehydrated under the sun until they obtain dry fruits with a humidity of 11–12%, and the shell is removed to obtain the green coffee beans (GC) [13]. The green beans are principally marketed on a small scale as mixtures of Typica, Bourbon, and Oro Azteca varieties, based on their color and size.

The next step is to determine the nutritional and chemical analyses of green and ground roasted beans that are currently marketed, in addition to the antioxidant effect, as well as the cup quality of coffee drinks.

### 2.1. Nutritional Composition

The bromatological analysis of green and processed beans in coffee allows for knowledge of the quality and nutritional composition of commercial coffees. The protein and carbohydrate content were similar between toasted beans and the mixture (GCM) of Typica, Bourbon, and Oro Azteca (Table 1); the fat content was higher in the dark roast coffee (DCR). The higher humidity content was found in GCM, which decreases as the roasting temperature is increased (MRC 180 °C and DCR 210 °C), while the ashes generated are similar in GCM and DRC samples. The humidity (<6.0%) and ash (<6.5%) contents in the medium roasting coffee (MRC) and DCR commercial coffee beans are within the range established by the Official Mexican Norm for roasted and ground coffee, NMX-F-013-SCFI-2010. The humidity of green beans is inferior to the range specified to prevent the growth of fungi and bacteria (10–12%) and preserve physical properties and nutritional content. Instead, only the fat content of DCR beans as an etheric extract (8–18%) achieves the parameter stipulated by the norm. The roasted coffee beans darken, and the released oils give them a shiny appearance.

In the *C. arabica* beans with light (176 °C), medium (204 °C), and dark (232 °C) roasts, the humidity content was reduced with the temperature increase, while protein (16%) and fat (16.2%) contents increased in the dark roast. Ashes and sugar contents were similar (2 °Brix) in the three roasts [16].

### 2.2. Chemical Analysis

The High-Performance Liquid Chromatography (HPLC) analysis showed that compounds of greater predominance identified in the infusions of greens and processed beans were CGA (λ = 330 nm) and caffeine (λ = 280 nm), with a retention time of 8.51 min and 8.88 min, respectively (Figure 2). Minority compounds around CGA and caffeine were detected at these long waves; one of them was caffeic acid with a retention time of 8.3 min. These compounds are related to the cup quality of the coffee drink, which gives it astringency and flavor [15,17].

The ^1^H and ^13^C nuclear magnetic resonance (NMR) analysis of the compound isolated from the MRC infusion validated that it corresponds to CGA; chemical displacements and coupling constants (Table 2) correspond to those reported in the literature for this compound [18].

Among the green beans of *C. arabica* varieties analyzed (Table 3), the Typica variety has the lowest contents of CGA (36.81 mg/g) and caffeine (1.16 mg/g). The GCM is composed of a greater proportion of the beans of the Typica species (Typica-Bourbon-Oro Azteca, 40–30–30%), and the content of both compounds in the GCM is close to those detected in this variety. It has been reported that in *C. arabica* green coffee, the CGA content ranges between 52 and 76 mg/g [19]. The CGA content in the Bourbon and Oro Azteca varieties is within this range.

When the green beans are subjected to the roasting process, the CGA content is reduced by the effects of temperature and time exposure. CGA is hydrolyzed into the molecules of simple phenols that compose it, caffeic acid and quinic acid [17]. In the GCM and MRC beans, the CGA content was similar, while in the DRC beans roasted at 210 °C, the CGA content was reduced by 53% in comparison to the unroasted beans (Table 3).

Studies carried out at different levels of roasting show that the content of CGA in the beans is correlated with the temperature and the time of roasting, finding greater content in a light roast (186.5 °C, 7:15 min) of 11.24 mg/g of coffee; when the roasting time is prolonged to a dark roast (186.5 °C, 14:02 min), the content is reduced by 70% [20].

Similarly, the Typica variety has a low content of caffeine (1.16 mg/g) compared with the other two varieties; therefore, the GCM presents a low level (0.87 mg/g) of caffeine (Table 3). The caffeine content determined in green beans increased in the roasted beans as the level of roasting increased (Table 3). Darkly roasted beans presented a higher caffeine content; when the green beans are roasted, they lose moisture, increase in size, and become porous, allowing a better caffeine extraction [21]. Nevertheless, this content is minor compared with that established for the commercialization of roasted coffee; according to the Mexican Norm (NMX-F-013-SCFI-2010), the acceptable caffeine content range is between 10 and 20 mg/g for roasted coffee.

The CGA and caffeine content are related to the variety of coffee, growing conditions, and degree of ripeness of the fruits. In a study conducted on coffees from Veracruz, Nayarit, Oaxaca, and Chiapas states of Mexico, it was found that the level of caffeine ranges between 2.9 and 7.0 mg/g of roasted coffee [22]. The caffeine content determined in the processed coffee from Guerrero State (Región de la Montaña) and reported in this study is between these ranges.

In Typica variety coffee beans from plants grown in two different places, the caffeine content in light roasting was 4.19 mg/g and 5.01 mg/g; when the level of roasting was increased to a dark level, the content presented an increase (5.18 mg/g and 6.12 mg/g), a similar effect that was reflected in this study [21]. Other studies on *C. arabica* have not reported variations in its content at different levels of roasting (11.9–13 mg/g), even using temperatures between 194 and 217 °C, showing a thermostable behavior of caffeine [23].

In other studies, it has been reported that in *C. arabica* beans from Brazil, the highest content of caffeine was present in light (6.42 mg/g) and medium (5.77 mg/g) roasts, compared with dark roast (2.63 mg/g). A similar effect was reported in the Typica and Bourbon varieties of coffee beans; the highest content was presented in light roasting (14.59 mg/g), followed by medium and dark (5.57 mg/g). The caffeine content in green beans of Sidama (16.4 mg/g), Yirgacheffe (15.72 mg/g), and Harar (15.03 mg/g) varieties was reduced to 7.96, 8.87, and 4.52 mg/g, respectively, after the beans were subjected to a dark roasting process [20,24].

### 2.3. Melanoidins

During roasting practice, the phenolic compounds can be degraded by Maillard and Strecker reactions, or they can be followed by the formation of new compounds such as melanoidins, acrylamides, and hydroxymethylfurfural [25,26]. Melanoidins give the beans a brown pigment, flavor, and color, and they are associated with antioxidant activity that is enhanced by simple phenolic compounds such as caffeic, ferulic, and chlorogenic acids binding to their structure [17].

In this study, two procedures were performed to determine the melanoidin content of the coffee infusions. In unclarified solutions, the melanoidins are present in the GCM, and the content more than doubled after beans were roasted in the DRC (~2.87 fold). The coffee beans presented a dark brown coloration (Table 2). Carrez I and II solutions are used to precipitate proteins and remove turbidity and micelles, reducing interference at the time of reading (λ = 420 nm). In the clarified samples, the melanoidin content was lower compared with those not clarified, but with similar behavior in their content; it was related to the increase in the level of roasting. The samples subjected to the clarification process lost color and presented the formation of a precipitate, which may be influencing the elimination of compounds of high molecular weight. The contents of melanoidins determined are lower than those reported in the literature (200–250 mg/g) [27,28]. Other authors report that the highest melanoidin content was determined in soluble coffee obtained from *C. canephora* (676 mg/g) and *C. arabica* varieties (peaberry, known as caracolillo in Spanish, 305 mg/g) roasted with sugar [28].

### 2.4. Sensory Evaluation

An important evaluation for commercial coffees is the cup quality of coffee drinks; this value depends on the physical and organoleptic properties of the beverage, such as flavor, aroma, acidity, body, and balance. These properties are associated with the content of their chemical compounds (Figure 1), mainly CGA, caffeine, caffeic acid, ferulic acid, vanillic acid, and cinnamic acid, among others [17]. In this work, CGA and caffeine contents were analyzed.

The GCM infusions had a pH of 5.6, the MRC infusions had a pH of 4.74, and the DRC infusions had a pH of 6.15. The MRC infusions had a higher acidity correlated with the CGA content (Table 3), as reported [16]. The sensory attributes established by the Mexican Norm PROY-NOM-255-SE-2021 in fragrance/aroma, flavor, acidity, balance, and taster score must present a score ≥ 8.0; in residual flavor and body, a score ≥ 7.5; and in uniformity, cup cleanliness, and sweetness, a score of 10. Natural coffees or honeyed coffees of specialty are those that present a total rating of 85 to 87.75 points according to the Mexican Norm.

The sensory analysis of the infusions prepared from MRC achieved a score of 86.25, and it was considered a natural specialty coffee. The points obtained in each parameter are within the values considered by the Norm (Table 4). In natural coffee or dry coffee, it is common to obtain characteristic fruity aromas and flavors due to the preservation of the peel and pulp of the fruit [29], such as those determined in this study, which present aromas of tropical fruits and flavors of white wine, grape, and honey.

The score of DRC coffee (84.25) was lower than that required to be considered a natural specialty coffee, but it is within the natural coffees named premium (80 to 84.75 points) according to the Norm, and attributes such as aroma, flavor, body, and balance presented the lowest score. In this drink, a floral and fruity aroma with date, grape, and red apple flavors was detected, along with tartaric citrus acidity, a juicy-silky body, and medium-high sweetness. The sensory analysis performed on the MRC beans is within the considered range of specialty coffees; a lower flavor was reported (7.75) and a greater body (8.25); in the case of DRC, beans showed a lower aroma (7.75), a lower balance (7.25), as well as a smaller body than in MRC beans.

In roasted coffee, it is very common to find mixtures of *C. arabica* varieties identifying different organoleptic characteristics influenced by the method to which the cherries are subjected to obtain the green beans. Other aspects that must be considered are the altitude at which the crops are located and their degree of maturity. The commercial coffees analyzed in this study were carefully collected by hand, considering the uniformity of ripe cherries, from *C. arabica* plant varieties grown from 1900 m.a.s.l. under the shade of other native trees of the place; furthermore, during drying, cherries were placed on beds to avoid contact with the soil and the growth of contaminating microorganisms. In addition, selected and sorted grains are stored in moisture-controlled spaces to preserve the quality of the green beans.

*C. arabica* beans from Brazil processed as natural coffee (87.8 points) presented better attributes in aroma, flavor, acidity, and body than honeyed coffee (83.8 points), possibly due to fermentation carried out by the microorganisms present in the fruit [30]. Likewise, natural coffee (dry, 80), wet (85), and semi-dry (86) with the fermentation of *C. arabica* variety Colombia presented similar attributes such as medium fruity body, medium fresh acidity, and chocolate and caramel flavors. Sensory attributes are related to geographic conditions, climate, altitude, and crop field practices [31].

### 2.5. Biological Activities

#### 2.5.1. Antioxidant Effect

The coffee beverage has antioxidant activity related to the content of chemical compounds such as caffeine, CGA, and melanoidins [17]. The antioxidant activity of processed coffee infusions was determined by the equivalents of Trolox and CGA related to the inhibitory effect of the radicals DPPH, ABTS, and FRAP (Table 5). In the tests carried out, the MRC extract showed a greater capacity for radical scavenging compared with the DRC extract. The CGA content in the infusion of MRC was related to the greater inhibitory activity, likely due to the low degradation of phenolic compounds involved in antioxidant processes. Furthermore, DRC presented a higher content of melanoidin, although its activity was lower than that presented by MRC. Finally, in the case of caffeine, it acts indirectly in the antioxidant process by increasing levels of glutathione [32]. Other molecules related to the main effect of coffee beverages are ferulic acid, caffeic acid, vanillic acid, guaiacol, epicatechin, catechin, and anthocyanins, which can unravel various mechanisms for the elimination or inhibition of free radicals [11,12,33].

The roasted beans of Colombia, Typica, and Bourbon varieties presented a similar effect to that reported in this study; the greatest antioxidant activity was found in the light and medium roasts through the tests carried out with DPPH and ABTS [20,31]. The high content of phenolic compounds provides greater radical inhibitory activity in beans with light roasting of the Cataui variety [34].

The concentration needed to inhibit 50% of DPPH and ABTS radicals depends on the content of phenolic compounds present, such as CGA. According to the tests performed, the MRC extract (higher CGA content, Table 3) presents a greater antioxidant effect (Table 5), and a lower concentration is required to achieve IC_50_ compared with DRC (Table 6). In *C. arabica*, the content of CGA was higher in green beans than in roasted beans, so less content of the green extract was required to achieve IC_50_ using the DPPH test [35].

#### 2.5.2. Cytotoxic Activity

Coffee, being a food product with high demand around the world, must be free of harmful chemical compounds and biological agents. It promotes the formation of toxic compounds in coffee beans when exposed to high temperatures, such as acrylamide and hydroxymethylfurfural, which are considered carcinogenic or genotoxic [26,36]. Based on the IC_50_ determined for DRC (216.26 ± 27.7 μg/mL) and MRC (234.63 ± 29.6 μg/mL) infusions on the growth of the 3T3-L1 fibroblast cell line at 48 h of exposition, it was determined that the infusions did not present cytotoxic effects. A plant extract has been determined to be cytotoxic when it presents IC_50_ values <100 μg/mL [37]. Studies on coffee extracts have reported a cytotoxic effect on prostate cancer cell lines DU145 and PC3, without showing toxic effects in macrophage cell lines (RAW 2647), hepatocytes (AML-12), or normal CCD-18Co fibroblast cells [34]. The commercial samples (MRC and DRC) analyzed did not show toxic effects, likely due to the low content of harmful compounds in these coffees.

The acrylamide content is regulated by each country; in the European Union, the permissible limit for acrylamide content is 400 μg/kg for roasted coffee and 800 μg/kg for soluble coffee. Studies reported variable amounts of acrylamide, but it has been shown that in dark coffees or with prolonged levels of roasting, the acrylamide content may be higher than in medium or light coffees [25,38].

Several studies have shown that the content of chemical compounds is influenced by the processing method of beans and infusion preparation [39]. In the infusion preparation, it is essential to consider the pressure, temperature, and contact time of the beans with the water; within the populations, the amount of coffee used can vary; for example, for filtered coffees in Europe, the United States, and Canada, 7 g per 100 mL of water is usually used, in Brazil 10 g, and in Italy 20 g [40]. In Mexico, it is common for coffee preparation to use one or two tablespoons per coffee cup for Turkish coffee or drip coffee. In this study, infusions were prepared with 6.6 g of coffee (equivalent to one tablespoon) per 100 mL of water by extraction using a French press due to its practicality.

On average, people usually ingest two to four cups (≈150 mL per cup) of coffee per day. Moderate coffee drinkers consume per day between 200 and 400 mg of caffeine and between 200 and 500 mg of CGA per cup [13,41]. Studies have shown that coffee consumption of four cups per day can benefit health by reducing the risk of Parkinson’s disease, Alzheimer’s disease, and type 2 diabetes mellitus [42]. Infusions of MCR using 9.9 g of coffee in 150 mL of water would contain 25.0 mg of caffeine and 299.5 mg of CGA, while in DRC infusions the content would be 38.4 mg of caffeine and 143.7 mg of CGA per cup. Coffee beverage consumption of MCR or DCR coffee can benefit people’s health.

The antioxidant properties of coffee infusions are mainly attributed to CGA and caffeine, although other compounds that influence antioxidant properties, such as epicatechin, catechin, and anthocyanins, have also been reported. Oxidative stress is a trigger for chronic degenerative diseases such as arthritis, diabetes mellitus, and cancer. Important pharmacological effects such as anti-inflammatory, regulation of glucose and lipid metabolisms, and anticancer have been reported for coffee beverages and CGA [13,15,40,43]. Caffeine exerts physiological effects associated with the functions of the central nervous system [44].

## 3. Materials and Methods

### 3.1. Biological Material

The cherries of *Coffea arabica* varieties Typica, Bourbon, and Oro Azteca employed in the present study were harvested in the 2020–2021 and 2021–2022 cycles in the plantations of the Cooperative Cafeticultores Mephaa “Region de La Montaña”, at the localities of La Soledad and Paraje Montero, municipality of Malinaltepec (longitude: 98.704167 and latitude: 17.164167), Guerrero, Mexico (Figure 3). Coffee cherries of each variety were dried on drying beds under the sun at room temperature for 15 days; after that time, the husk was removed to obtain the GC.

Subsequently, the GC of the Typica, Bourbon, and Oro Azteca varieties were mixed (GCM) in a ratio of 40–30–30%. The mixture of coffee beans was processed in a 100MEX^®^ brand industrial roaster with a steel rotating cylinder to reach an Agtron Gourmet Bean roasting level of 45 (180 °C per 15 min) to obtain a medium roast coffee (MRC) and an Agtron Gourmet Bean roasting level of 35 (210 °C per 15 min) for a dark roast coffee (DRC). The roasted beans are transferred onto plates and allowed to cool at room temperature (Figure 4).

### 3.2. Bromatological Analysis

The analysis of the nutritional components of GCM, MRC, and DRC beans was performed based on the standard methods of the American Cereal Chemical Association (AACC International) [45]. The methods employed were moisture (44–15.02), protein (46–16.01), fat (30–25.01), and ash content values (08–01.01). Moisture (%) was determined by the weight difference of the grain before and after drying in a forced-air oven (VWR, DHG-9070A)^®^ (Beijing, China) for 5 min. The ash content (%) was obtained from the difference in weight of the waste generated by the samples placed in a porcelain pot and incinerated in a muffle at 550 °C for 4 h. The fats of the grains were extracted by Soxhlet with petroleum ether, the solvent was evaporated, and the extracted residue was kiln-dried at 100 °C, weighed, and expressed as % ethereal extract (crude fat). The protein content (%) was determined by the quantification of total free nitrogen by the modified Kjeldahl method. Carbohydrate content was calculated by subtracting the sum of the percentages of moisture, lipids, protein, and ash from 100%.

The values of humidity, ash, proteins, lipids, and carbohydrates obtained in the GCM, MRC, and DRC were expressed as an average of three replicates and their standard deviation (SD). Each variable was analyzed with a simple ANOVA and a Tukey *post*-test with a confidence level of 95% (*p* < 0.05) using the SAS System for Windows 9.1 software (Statistical software, SAS Institute, Inc., Cary, NC, USA).

### 3.3. Chemical Analyses

#### 3.3.1. Infusion Preparation

The beans of Typica-GC, Bourbon-GC, Oro Azteca-GC, GCM, MRC, and DRC were ground in an MCT-750 100MEX^®^ (Veracruz, Mexico) grinder with a particle size of 1.0 mm. Infusions of each coffee were prepared in the laboratory in triplicate at room temperature with 13.2 g of ground coffee beans in 200 mL of boiling water to 98 °C in a French press for 5 min. The infusions were filtered, and the pH was measured with a potentiometer Oakton pH 510; subsequently, the infusions were concentrated at reduced pressure in a rotatory evaporator (Heidolph Laborota 4000). Then, the extracts were freeze-dried (Heto Drywinner DW3), and the powders were stored in amber glass containers at room temperature.

#### 3.3.2. Chlorogenic Acid and Caffeine Quantification

The concentrations of chlorogenic acid (CGA) and caffeine (CAF) in the Typica-GC, Bourbon-GC, Oro Azteca-GC, GCM, MRC, and DCR infusions were determined by High-Performance Liquid Chromatography (HPLC) in Waters equipment consisting of a separation module (Waters 2695) and a photodiode detector (Waters 2696). For this purpose, 10 μL of each infusion at concentrations of 0.25, 0.5, and 1.0 mg/mL were injected and eluted through an RP-18 column (250 × 4.6 mm, 5 µm, SUPELCO Discovery^®^, Merck, Darmstadt, Germany) with a flow of 0.9 mL/min in a gradient system of 30 min based on water HPLC grade (VWR, Mississauga, ON, Canada) with 0.5% trifluoroacetic acid (Sigma-Aldrich, Saint Louis, MO, USA) (A) and acetonitrile-HPLC (Merck, Germany) (B). Gradient flow starts at A-100%, with gradient changes to 95% in 2 min, 70% in 2 min, 50% in 17 min, 20% in 3 min, 0% A and 100% B in 3 min, and finally system return to initial conditions (A-100%) in 3 min. Data were processed with the Empower Pro 3.0 software (Waters, Milford, MA, USA), and the chromatograms were obtained at wavelengths (λ) of 330 nm for CGA and at λ = 280 nm for caffeine. The identification of both compounds was determined by comparison of their retention time (RT) and absorption spectra. The CGA and caffeine concentrations were calculated according to external standards for CGA (3-(3,4-Dihydroxycinnamoyl) quinic acid; ≥95% purity, Sigma-Aldrich) and caffeine (≥99% purity, Sigma-Aldrich). The calibration curves of CGA and caffeine were constructed using a lineal square model y=mx+b with the Microsoft Office Excel 365 Software (Microsoft^®^ Excel V.16.70) with correlation coefficients ≥0.9995; for CGA, it was a regression equation of y = 11,702x + 19,276 with an R^2^ = 0.9982, and for caffeine, y = 87,483x + 38,786 with an R^2^ = 0.9993.

The CGA and CAF contents were expressed in mg/g of coffee beans as the mean of nine analyses and their standard deviation (SD). The GC contents of the varieties, GCM, MRC, and DRC infusions, were compared with an ANOVA and a Tukey post-test with a 95% confidence level (*p* < 0.05).

#### 3.3.3. Chemical Fractionation of MRC Infusion

The MRC extract (2 g) was fractionated by open column chromatography (2.7 cm diameter × 44 cm high), packed with 20 g of RP-18 silica gel (Supelco, Germany), and eluted with a gradient system of H_2_O: CH_3_CN. Aliquots of 10 mL were collected with an initial system of 100% H_2_O (1–10) and polarity changes from CH_3_CN at 5% (11–15), CH_3_CN at 50% (16–17), and CH_3_OH at 100% (18). The fractions were analyzed by reverse-phase-thin layer chromatography with an elution system of 90:10 H_2_O:CH_3_CN and displayed in a UV lamp (UVP UVGL-58) at λ = 254 nm and λ = 365 nm. Aliquots with a similar chromatographic profile (13–15) were grouped and diluted in dimethyl sulfoxide to be analyzed by NMR of ^1^H and ^13^C spectra at 100 MHz, 2-dimensional (2-D) correlated spectroscopy (COSY), heteronuclear simple quantum coherence (HSQC), and heteronuclear multiple bond coherence (HMBC) at 400 MHz on Varian INOVA-400 equipment.

### 3.4. Melanoidins

The melanoidin content was analyzed in infusions of GCM, MRC, and DRC by two different procedures:(1)Serial dilutions (2.0–0.0625 mg/mL) were prepared from a solution of 10 mg/mL of each infusion, and the absorbance of each concentration was measured at λ_max_ = 420 nm in a UV-VIS spectrophotometer (Genesys 20-Thermo Scientific, Waltham, MA, USA). The melanoidin content in the infusions was determined by the Lambert-Beer formula: C = A/cb, where C is concentration, A absorbance, b cell length (1 cm), and c extinction coefficient (1.1289 L/g cm) [28,46].(2)Extracts from 2 g of each infusion were dissolved in 20 mL distilled water, and they were filtrated through Acrodiscs Pall^®^ (0.45 μm). The calibration curves for each coffee were built from dilutions with absorbances between 1.0 and 0.01. For melanoidin determination, 1 mL of the filtrate was diluted with water (1/5, *v*/*v*), and 1 mL of Carrez I and II solutions were added (Sigma-Aldrich). The solution was homogenized and completed to a volume of 10 mL. Then, each sample was centrifuged at 4000 rpm for 5 min, and the clarified samples were filtered through Acrodiscs Pall^®^ (Pall Port Washington, NY, USA) (0.20 μm). The corresponding readings for melanoidins were carried out to obtain the content and the specific extinction coefficient (K_mix_) determined by Lambert-Beer’s law.

The melanoidin content values in the GCM, MRC, and DRC were expressed as the mean of three analyses and their SD, compared with an ANOVA and a Tukey *post*-test with a 95% confidence level (*p* < 0.05).

### 3.5. Sensory Assessment

Infusions in cups of MRC or DRC ground coffee (MCT-750 100MEX^®^ grinder (100MEX Veracruz, Mexico) with a particle size of 1.0 mm) were prepared with 8.25 g in 150 mL of distilled water at 93 °C in a simple infusion. The sensory evaluation of the coffee in cup was carried out by four tasters certified by the Mexican Association of Specialty Coffees and Cafeterias, AC (AMCCE, abbreviations in Spanish), following the cupping protocol of the Specialty Coffees of America Association (SCAA) [47]. The parameters of the SCAA are based on a reference of 100 points; a score of 0 to 10 points was given to each organoleptic characteristic of aroma/fragrance, flavor, residual flavor, sweetness, acidity, body, uniformity, balance, clean cup, and taster. The point total sum was the result of the sensory analysis; scores ≥ 80 are considered very good coffees and are classified as specialty coffees.

### 3.6. Biological Analyses

#### 3.6.1. Antioxidant Activity Assays

##### Radical Scavenging of 2,2-Diphenyl-1-picrylhydrazyl (DPPH)

In each well of microplates of 96 wells, 175 μL of DPPH (1 mg/mL of methanol) solution was added, followed by 25 μL of CGA (3.0–100 µg/mL), Trolox (6-Methoxy-2,5,7,8-tetramethylchromane-2-carboxylic acid) (3.0–100 µg/mL), MRC (0.1–5.0 mg/mL), or DRC (0.1–5.0 mg/mL) solutions. The absorbance was measured at the beginning of the reaction (*A*_0_), and the plates were stored protected from light for 30 min. After this period, the absorbance was measured (*A*_1_) again with an ELISA lector (Perkin-Elmer Lambda 40 UV/Vis) to λ = 515 nm. The inhibition percentage was calculated with the equation:$$\%\;inhibition=\left[\frac{\left(A_{0}-A_{1}\right)}{A_{0}}\right]\times 100$$

The CGA, Trolox, MRC, and DRC mean inhibitory concentrations (IC_50_) of scavenging DPPH radicals were determined based on the curves generated by recording the inhibition percentages of the reaction against the concentration. In addition, the absorbance values of the samples were compared with the graphed curves of the inhibition percentage against the concentration of the standards and reported in GCA or Trolox equivalents (eq CGA and eq Trolox) [25].

The IC_50_ and GCA, or Trolox equivalents, were expressed as the mean of three replicates and their SD. Each variable of MRC and DRC was compared by a Student’s *t*-test with a *p* < 0.05.

##### ABTS (2,2′-Azinobis (3-Ethylbenzothiazoline-6 sulfonic Acid) Radical

The 7 mM ABTS radical solution was prepared with persulfate of potassium 140 μM in the dark under agitation for 12 to 16 h; after, the solution was diluted with methanol until it had an absorbance value of 0.7 at λ = 734 nm. In each well, 230 μL of ABTS solution and 20 μL of CGA (0.3–1.3 µg/mL), Trolox (0.3–1.3 µg/mL), MRC (0.1–5.0 mg/mL), or DCR (5.0–0.1 mg/mL) were added. After a 30 sec stirring, the absorbance was determined at λ = 734 nm. The antioxidant activity was expressed by inhibition percentage, IC_50,_ and CGA or Trolox equivalents, as previously described.

The IC_50_ and GCA, or Trolox equivalents, were expressed as the mean of three replicates and their SD. Each variable of MRC and DRC was compared by a Student’s *t*-test with a *p* ≤ 0.05 value.

##### Ferric Reducing/Antioxidant Power (FRAP)

The FRAP solution was produced with 1 mL of 10 mM 2,4,6-Tris(2-pyridyl)-s-triazine (TPTZ) solution and 1 mL 20 mM ferric chloride (FeCl_3_^.^6H_2_O) solution in 10 mL of 300 mM sodium acetate buffer (CH_3_COONa) to pH 3.6. The FRAP solution was prepared freshly and kept at 37 °C. To each well, 175 μL of FRAP was added, along with 50 μL of CGA (13–0.3 µg/mL), Trolox (0.3–1.3 µg/mL), MRC (0.1–5.0 mg/mL), or DCR (0.1–5.0 mg/mL). The absorbance was measured at λ = 595 nm after 30 s of the reaction.

The IC_50_ and GCA, Trolox, or ferric sulfate equivalents were expressed as the mean of three replicates and their SD. Each variable of MRC and DRC was compared by a Student’s *t*-test with a *p* ≤ 0.05 value.

#### 3.6.2. Cytotoxic Evaluation

The 3T3-L1 (ATCC^®^ CL-173) cell line was used for cytotoxic assays; this culture is a mouse embryonic fibroblast cell line obtained from IN VITRO S.A. The 3T3-L1 fibroblasts are committed to differentiation into adipocytes. The cells were cultivated in Dubelco’s Modified Eagle medium (DMEM) supplemented with 10% fetal bovine serum, L-glutamine (1 mL × 100), antibiotic 3X (1 mix100), non-essential amino acids (1 mL × 100), and sodium bicarbonate (3 mL × 100). Cellular cultures were incubated at 37 °C in a humidified atmosphere with 5% CO_2_.

For the cytotoxic assay, the 3T3-L1 cells were cultivated in plates of 96 wells with a cellular density of 3 × 10^4^ cells per well. After 24 h of plate attachment, the non-adherent cells were eliminated and subsequently treated with DMSO 1% as the negative control, different concentrations of MRC and DRC infusions (15.6–500 μg/mL), and paclitaxel as the positive control (0.6–20 μg/mL). The plates were incubated for 24 and 48 h at the conditions above described.

After each incubation time, the medium was discarded, and 80 μL of DMEM medium and 20 μL of (3-(4,5-dimethylthiazol-2-yl)-2,5-diphenyltetrazolium bromide) (MTT, 5 mg/mL in PBS) were added to each well and incubated for 4 h at 37 °C to allow the formation of formazan crystals. The supernatant was removed, 100 μL of isopropanol was added, and the plates were incubated at room temperature with stirring for 15 min until the formazan crystals were dissolved. The optical density of the solution was measured with a microplate spectrophotometer at λ = 490 nm.

The cell viability percentage was determined by the following equation:$$Cell\;viability(\%)=\left[\frac{\left(negative\;control\;OD-tested\;sample\;OD\right)}{negative\;contro\;OD}\right]\times 100$$

The mean inhibitory concentration (IC_50_) for each infusion was calculated with a dose-response curve regression analysis.

## 4. Conclusions

The artisanal coffees showed variable chemical contents related to the roasting process to which they were subjected. The nutritional composition of MRC and DRC beans is according to normative requirements. Both infusions presented variations in sensory attributes, allowing the classification of the MRC as specialty coffee and the DRC coffee as premium. It is well known that coffee consumption is related to its flavor and aroma; both coffees had aromas of tropical fruits and flavors of white wine, grape, and honey.

The conditions of cultivation, the proportions of the *C. arabica* varieties in the coffee bean mixture, the roosting processing of the beans, and the infusion preparation are parameters related to the CGA and caffeine contents. CGA content was reduced by the effect of roasting, while caffeine and melanoidins were increased, although below the suggested levels by the norm. The commercial samples analyzed showed antioxidant activity without presenting toxic effects; the content of harmful compounds in these coffees may be at low concentrations.

The results obtained in this experimental work will be the basis for making modifications to improve the manufacturing processes of these commercial coffees, comply with the standards for green and roasted coffee dealing by the Mexican Standard, and preserve their chemical composition and biological potential.

## Figures and Tables

**Figure 1 molecules-28-04685-f001:**
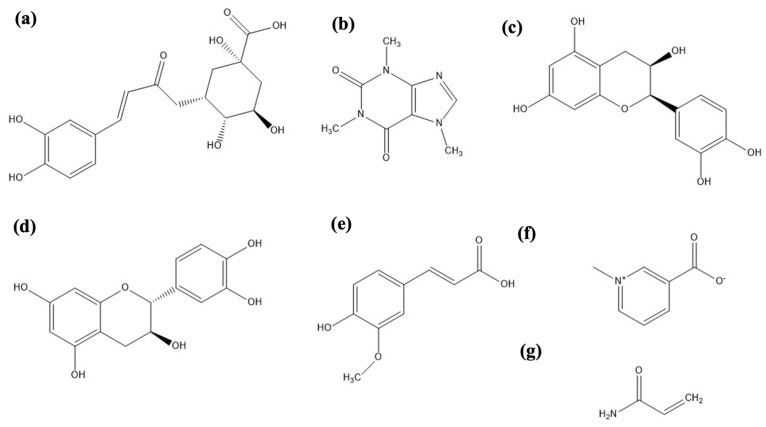
Chemical structures of main compounds identified in the beans of *Coffea arabica*: (**a**) 3-*O*-caffeoylquinic acid (CGA); (**b**) Caffeine; (**c**) Epicatechin; (**d**) Catechin; (**e**) Ferulic acid; (**f**) Trigonelline; (**g**) Acrylamide.

**Figure 2 molecules-28-04685-f002:**
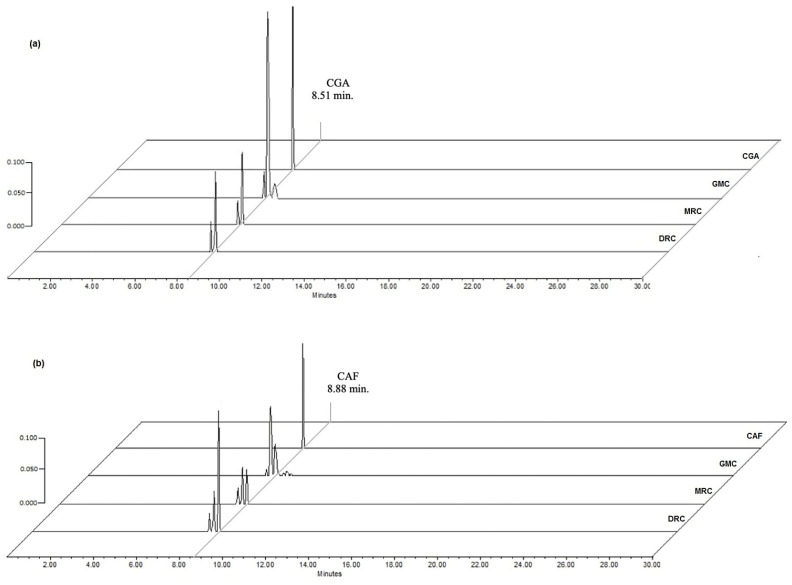
Chromatograms of HPLC at λ = 330 nm of (**a**) CGA standard (25 µg/mL) and infusions of GCM (125 µg/mL), MCR, and DCR (500 µg/mL); (**b**) at λ = 280 nm of caffeine (CAF) standard (10 µg/mL) and infusions of GCM (125 µg/mL), MCR, and DCR (500 µg/mL).

**Figure 3 molecules-28-04685-f003:**
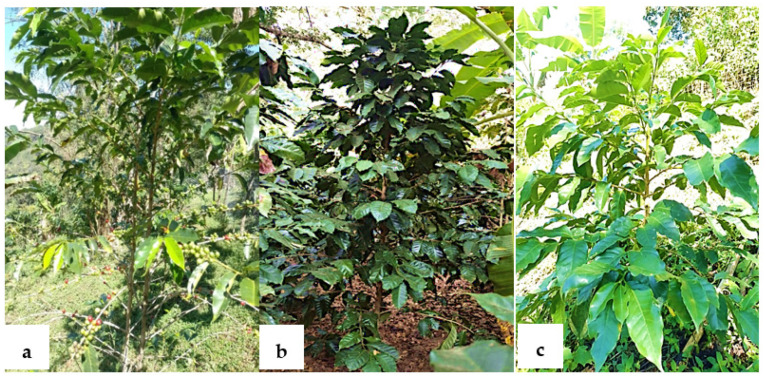
Photographs of the varieties *Coffea arabica* (**a**) Typica, (**b**) Bourbon, and (**c**) Oro Azteca grew in the Paraje Montero locality municipality of Malinaltepec at 1980 m.a.s.l.

**Figure 4 molecules-28-04685-f004:**
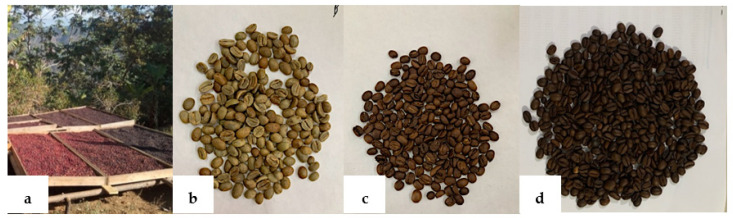
Processed cherries and beans of *C. arabica* (**a**) Drying of cherries of different varieties (GCM); (**b**) Mixture of green beans of different varieties; (**c**) Medium-roasted coffee beans (MRC); (**d**) Dark-roasted coffee beans (DRC).

**Table 1 molecules-28-04685-t001:** Nutritional composition of commercial coffee with *Coffea arabica* varieties in the mixture.

Coffee	Content in Percentage (%)				
	Humidity	Ash	Fats	Proteins	Carbohydrates
GCM	8.48 ± 0.13 **	4.54 ± 0.06 **	5.09 ± 0.89	12.34 ± 0.29	69.56 ± 1.06
MRC	4.23 ± 0.14 *	3.84 ± 0.11	6.48 ± 0.34 *	13.04 ± 0.28	72.41 ± 1.26
DRC	3.59 ± 0.12	4.44 ± 0.08 **	8.15 ± 0.63 **	13.01 ± 0.38	70.81 ± 1.11

Values are mean ± standard deviation (*n* = 3). According to the ANOVA and Tukey’s test, the means with * and ** were significantly different. Humidity F = 1251.75 *p* ≤ 0.0001, Tukey_0.05_ = 0.33; Ash F = 62.89 *p* ≤ 0.0001, Tukey_0.05_ = 0.20; Fats F = 16.26, *p* ≤ 0.0001, Tukey_0.05_ = 1.65; Proteins F = 4.56 *p* > 0.05, Tukey_0.05_ = 0.80; Carbohydrates F = 7.55 *p* ≤ 0.0001; Tukey_0.05_ = 2.25.

**Table 2 molecules-28-04685-t002:** ^1^H (400 MHz) and ^13^C NMR (100 MHz) Spectroscopic Data of Chlorogenic Acid (MeOH-d_4_, δ, ppm, J/Hz).

			Data of [18]	
C Atom	δ ^1^H-Experimental	δ ^13^C-Experimental	δ ^1^H	δ ^13^C
1		70.24		71.06
2a b	2.09 (m) 1.92 (dd,12.1, 12.4 Hz)	39.44	2.21 (m)	37.65
3	5.23 (ddd, 5.1,5.5, 10.2 Hz)	71.29	5.17	71.06
4	3.62 (d, br, 10.1 Hz)	72.97	4.89	68.48
5	4.12 (s, br)	71.91	4.77	73.90
6a b	1.98 (m) 1.98 (m)	37.21	1.84	36.66
1′		126.03		126.05
2′	7.10 (d,1.4 Hz)	115.06	7.00	114.99
3′		146.07		148.80
4′		148.80		145.71
5′	6.82 (d,8.1 Hz)	116.25	6.98	116.20
6′	7.03 (dd,1.5, 8.1 Hz)	121.68	7.00	114.99
7′	7.52 (d,15.8 Hz)	145.18	7.42	145.71
8′	6.30 (d,15.9 Hz)	115.20	6.15	114.99
9′		166.74		166.18
COOH		175.88		175.38

**Table 3 molecules-28-04685-t003:** Contents of CGA, caffeine, and melanoidins in green and processed beans of *Coffea arabica* varieties.

Coffee Beans	CGA	Caffeine	Melanoidins		
			Unclarified	Clarified	
	mg/g Coffee				K_mix_ Lg^−1^cm^−1^
GCM	30.81 ± 2.22	0.87 ± 0.09	15.41 ± 1.15	2.04 ± 0.88	0.07
Bourbon-GC	55.75 ± 2.31 **	1.78 ± 0.12 **	-	-	-
Oro Azteca-GC	54.63 ± 2.43 **	1.77 ± 0.15 **	-	-	-
Typica-GC	36.81 ± 0.10	1.16 ± 0.18	-	-	-
MRC	30.26 ± 0.45 **	2.52 ± 0.17 *	85.51 ± 5.99 *	18.95 ± 1.9 **	1.586
DRC	14.52 ± 0.65	3.88 ± 0.23 **	96.79 ± 3.44 **	29.06 ± 7.7 **	1.614

Values are mean ± standard deviation (*n* = 9). According to the ANOVA and Tukey’s test, the means with ** were significantly different. CGA F = 328.13 *p* ≤ 0.0001, Tukey_0.05_ = 2.66; caffeine F = 101.83 *p* ≤ 0.0001, Tukey_0.05_ = 0.17 in mg/g of coffee in green beans; CGA F = 415.72 *p* ≤ 0.0001, Tukey_0.05_ = 1.60; caffeine F = 678.07 *p* ≤ 0.0001, Tukey_0.05_ = 0.20 in mg/g of coffee in processed beans. For melanoidins, values are mean ± standard deviation (*n* = 3). According to the ANOVA and Tukey’s test, the means with * and ** were significantly different. Unclarified melanoidins F = 1423.09 *p* ≤ 0.0001, Tukey_0.05_ = 4.055 in mg/g of coffee, and clarified melanoidins F = 26.45 *p* ≤ 0.0001, Tukey_0.05_ = 11.516 in mg/g of coffee.

**Table 4 molecules-28-04685-t004:** Sensory profile of commercial coffee blend of *Coffea arabica* varieties.

Coffee Beans	MRC	DRC
Aroma	8.00 ± 0.16 *	7.75 ± 0.20
Taste	7.75 ± 0.29	7.75 ± 0.29
Aftertaste	8.00 ± 0.20	8.00 ± 0.20
Acidity	8.00 ± 0.61	8.00 ± 0.13
Body	8.25 ± 0.20 *	8.00 ± 0.13
Balance	8.00 ± 0.29 *	7.25 ± 0.29
Uniformity	10 ± 0	10 ± 0
Clean cup	10 ± 0	10 ± 0
Sweetness	10 ± 0	10 ± 0
Taster score	8.25 ± 0.29 *	7.5 ± 0.41
Total Score	86.25	84.25

Values are mean ± standard deviation (*n* = 4). According to the Student’s *t*-test (*p* ≤ 0.05), the means with * were significantly different.

**Table 5 molecules-28-04685-t005:** Antioxidant activity of commercial coffee with *Coffea arabica* varieties in the mixture.

Sample	DPPH		ABTS		FRAP		
	eq CGA	eq Trolox	eq CGA	eq Trolox	eq CGA	eq Trolox	eq FeSO4
MRC	1.60 ± 0.27 *	52.74 ± 4.84 *	16.09 ± 0.33 *	14.39 ± 1.16	16.22 ± 1.04 *	14.59 ± 2.35 *	54.68 ± 1.46 *
DRC	1.12 ± 0.37	42.52 ± 1.91	12.49 ± 0.46	12.15 ± 0.49	8.82 ± 0.94	6.38 ± 1.40	33.30 ± 0.63

Values are mean ± standard deviation (*n* = 3). According to the Student’s *t*-test (*p* ≤ 0.05), the means with * were significantly different.

**Table 6 molecules-28-04685-t006:** Inhibitory concentration (IC_50_) of commercial coffee with *Coffea arabica* mixture varieties.

Assay	IC_50_					
	MRC	DRC	MRC	DRC	CGA	Trolox
	mg/mL Extract		μg/mL CGA Content		μg/mL Standard	
DPPH	2.22 ± 0.08	2.59 ± 0.05 *	56.92 ± 1.90	66.20 ± 1.46 *	28.18 ± 0.83	91.88 ± 3.75 *
ABTS	0.38 ± 0.02	0.49 ± 0.02 *	9.69 ± 0.35	12.67 ± 0.44 *	6.51 ± 0.16 *	6.29 ± 0.03

Values are mean ± standard deviation (*n* = 3). According to the Student’s *t*-test (*p* ≤ 0.05), the means with * were significantly different.

## Data Availability

All data generated during this study are included in this published article.

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
