# Peer review of "Chemical and Biological Characterization of Green and Processed Coffee Beans from *Coffea arabica* Varieties"

_molecules, 2023, doi:10.3390/molecules28124685_

Round 1

Reviewer 1 Report

The manuscript molecules-2422094 entitled: “Chemical and biological characterization of green and processed coffee beans from Coffea arabica varieties” deals with the examination of coffee quality (physical, chemical, and biological characterics).

I think the work is well conceived, interesting, but not well presented. I suggest that the authors edit the text, paying particular attention to the results and discussion, so that one can understand the significance of the obtained results. English must be improved!

Authors must immerse themselves in their work and present the results in a meaningful way. It is very confusingly written. Many sentences are too long. In many places, it is not clear why the data from the literature are cited, they are not connected with the obtained results.

-A comment related to Figure 2: is it possible to have only one compound at 330 nm, and one at 280, and that the rest of the baseline is flat, there are no smaller peaks besides the dominant ones?

-I don't see that there is any diagram or table showing the results and the dependence of the content of caffeine and chlorogenic acid depending on % roasted.

In my opinion, the current form of the work does not satisfy the criteria to be published. I suggest that the paper goes through a major review and then return to the reviewer for reading.

English must be imroved.

Author Response

  1. Authors must immerse themselves in their work and present the results in a meaningful way. It is very confusingly written. Many sentences are too long. In many places, it is not clear why the data from the literature are cited, they are not connected with the obtained results.

All the manuscript was reviewed by the authors,

  1. A comment related to Figure 2: is it possible to have only one compound at 330 nm, and one at 280, and that the rest of the baseline is flat, there are no smaller peaks besides the dominant ones?

In the figure 2, the chromatograms of chlorogenic acid standard (CGA) to λmax = 320 nm and of caffeine (CAF) to λmax = 280 nm shown only one compound; to these long waves, it was possible to see two or three compounds around of them, mainly for the green coffee infusion.

  1. I don't see that there is any diagram or table showing the results and the dependence of the content of caffeine and chlorogenic acid depending on % roasted.

The contents of chlorogenic acid, caffeine and melanoidins in the green beans of C. arabica varieties and the coffee bean mixture, and medium roasted coffee (MRC) and  dark rosted coffee (DRC) are included in the table 3.

In my opinion, the current form of the work does not satisfy the criteria to be published. I suggest that the paper goes through a major review and then return to the reviewer for reading.

The manuscript molecules-2422094 entitled: “Chemical and biological characterization of green and processed coffee beans from Coffea arabica varieties” deals with the examination of coffee quality (physical, chemical, and biological characteristics).

I think the work is well conceived, interesting, but not well presented. I suggest that the authors edit the text, paying particular attention to the results and discussion, so that one can understand the significance of the obtained results. English must be improved!

  1. Authors must immerse themselves in their work and present the results in a meaningful way. It is very confusingly written. Many sentences are too long. In many places, it is not clear why the data from the literature are cited, they are not connected with the obtained results.

All the manuscript was reviewed by the authors,

  1. A comment related to Figure 2: is it possible to have only one compound at 330 nm, and one at 280, and that the rest of the baseline is flat, there are no smaller peaks besides the dominant ones?

In the figure 2, the chromatograms of chlorogenic acid standard (CGA) to λmax = 320 nm and of caffeine (CAF) to λmax = 280 nm shown only one compound; to these long waves, it was possible to see two or three compounds around of them, mainly for the green coffee infusion.

  1. I don't see that there is any diagram or table showing the results and the dependence of the content of caffeine and chlorogenic acid depending on % roasted.

The contents of chlorogenic acid, caffeine and melanoidins in the green beans of C. arabica varieties and the coffee bean mixture, and medium roasted coffee (MRC) and  dark rosted coffee (DRC) are included in the table 3.

In my opinion, the current form of the work does not satisfy the criteria to be published. I suggest that the paper goes through a major review and then return to the reviewer for reading.

The manuscript molecules-2422094 entitled: “Chemical and biological characterization of green and processed coffee beans from Coffea arabica varieties” deals with the examination of coffee quality (physical, chemical, and biological characteristics).

I think the work is well conceived, interesting, but not well presented. I suggest that the authors edit the text, paying particular attention to the results and discussion, so that one can understand the significance of the obtained results. English must be improved!

  1. Authors must immerse themselves in their work and present the results in a meaningful way. It is very confusingly written. Many sentences are too long. In many places, it is not clear why the data from the literature are cited, they are not connected with the obtained results.

All the manuscript was reviewed by the authors,

  1. A comment related to Figure 2: is it possible to have only one compound at 330 nm, and one at 280, and that the rest of the baseline is flat, there are no smaller peaks besides the dominant ones?

In the figure 2, the chromatograms of chlorogenic acid standard (CGA) to λmax = 320 nm and of caffeine (CAF) to λmax = 280 nm shown only one compound; to these long waves, it was possible to see two or three compounds around of them, mainly for the green coffee infusion.

  1. I don't see that there is any diagram or table showing the results and the dependence of the content of caffeine and chlorogenic acid depending on % roasted.

The contents of chlorogenic acid, caffeine and melanoidins in the green beans of C. arabica varieties and the coffee bean mixture, and medium roasted coffee (MRC) and  dark rosted coffee (DRC) are included in the table 3.

In my opinion, the current form of the work does not satisfy the criteria to be published. I suggest that the paper goes through a major review and then return to the reviewer for reading.

The manuscript molecules-2422094 entitled: “Chemical and biological characterization of green and processed coffee beans from Coffea arabica varieties” deals with the examination of coffee quality (physical, chemical, and biological characteristics).

I think the work is well conceived, interesting, but not well presented. I suggest that the authors edit the text, paying particular attention to the results and discussion, so that one can understand the significance of the obtained results. English must be improved!

  1. Authors must immerse themselves in their work and present the results in a meaningful way. It is very confusingly written. Many sentences are too long. In many places, it is not clear why the data from the literature are cited, they are not connected with the obtained results.

All the manuscript was reviewed by the authors,

  1. A comment related to Figure 2: is it possible to have only one compound at 330 nm, and one at 280, and that the rest of the baseline is flat, there are no smaller peaks besides the dominant ones?

In the figure 2, the chromatograms of chlorogenic acid standard (CGA) to λmax = 320 nm and of caffeine (CAF) to λmax = 280 nm shown only one compound; to these long waves, it was possible to see two or three compounds around of them, mainly for the green coffee infusion.

  1. I don't see that there is any diagram or table showing the results and the dependence of the content of caffeine and chlorogenic acid depending on % roasted.

The contents of chlorogenic acid, caffeine and melanoidins in the green beans of C. arabica varieties and the coffee bean mixture, and medium roasted coffee (MRC) and  dark rosted coffee (DRC) are included in the table 3.

In my opinion, the current form of the work does not satisfy the criteria to be published. I suggest that the paper goes through a major review and then return to the reviewer for reading.

The manuscript molecules-2422094 entitled: “Chemical and biological characterization of green and processed coffee beans from Coffea arabica varieties” deals with the examination of coffee quality (physical, chemical, and biological characteristics).

I think the work is well conceived, interesting, but not well presented. I suggest that the authors edit the text, paying particular attention to the results and discussion, so that one can understand the significance of the obtained results. English must be improved!

  1. Authors must immerse themselves in their work and present the results in a meaningful way. It is very confusingly written. Many sentences are too long. In many places, it is not clear why the data from the literature are cited, they are not connected with the obtained results.

All the manuscript was reviewed by the authors,

  1. A comment related to Figure 2: is it possible to have only one compound at 330 nm, and one at 280, and that the rest of the baseline is flat, there are no smaller peaks besides the dominant ones?

In the figure 2, the chromatograms of chlorogenic acid standard (CGA) to λmax = 320 nm and of caffeine (CAF) to λmax = 280 nm shown only one compound; to these long waves, it was possible to see two or three compounds around of them, mainly for the green coffee infusion.

  1. I don't see that there is any diagram or table showing the results and the dependence of the content of caffeine and chlorogenic acid depending on % roasted.

The contents of chlorogenic acid, caffeine and melanoidins in the green beans of C. arabica varieties and the coffee bean mixture, and medium roasted coffee (MRC) and  dark rosted coffee (DRC) are included in the table 3.

In my opinion, the current form of the work does not satisfy the criteria to be published. I suggest that the paper goes through a major review and then return to the reviewer for reading.

The manuscript molecules-2422094 entitled: “Chemical and biological characterization of green and processed coffee beans from Coffea arabica varieties” deals with the examination of coffee quality (physical, chemical, and biological characteristics).

I think the work is well conceived, interesting, but not well presented. I suggest that the authors edit the text, paying particular attention to the results and discussion, so that one can understand the significance of the obtained results. English must be improved!

  1. Authors must immerse themselves in their work and present the results in a meaningful way. It is very confusingly written. Many sentences are too long. In many places, it is not clear why the data from the literature are cited, they are not connected with the obtained results.

All the manuscript was reviewed by the authors,

  1. A comment related to Figure 2: is it possible to have only one compound at 330 nm, and one at 280, and that the rest of the baseline is flat, there are no smaller peaks besides the dominant ones?

In the figure 2, the chromatograms of chlorogenic acid standard (CGA) to λmax = 320 nm and of caffeine (CAF) to λmax = 280 nm shown only one compound; to these long waves, it was possible to see two or three compounds around of them, mainly for the green coffee infusion.

  1. I don't see that there is any diagram or table showing the results and the dependence of the content of caffeine and chlorogenic acid depending on % roasted.

The contents of chlorogenic acid, caffeine and melanoidins in the green beans of C. arabica varieties and the coffee bean mixture, and medium roasted coffee (MRC) and  dark rosted coffee (DRC) are included in the table 3.

In my opinion, the current form of the work does not satisfy the criteria to be published. I suggest that the paper goes through a major review and then return to the reviewer for reading.

The manuscript molecules-2422094 entitled: “Chemical and biological characterization of green and processed coffee beans from Coffea arabica varieties” deals with the examination of coffee quality (physical, chemical, and biological characteristics).

I think the work is well conceived, interesting, but not well presented. I suggest that the authors edit the text, paying particular attention to the results and discussion, so that one can understand the significance of the obtained results. English must be improved!

  1. Authors must immerse themselves in their work and present the results in a meaningful way. It is very confusingly written. Many sentences are too long. In many places, it is not clear why the data from the literature are cited, they are not connected with the obtained results.

All the manuscript was reviewed by the authors,

  1. A comment related to Figure 2: is it possible to have only one compound at 330 nm, and one at 280, and that the rest of the baseline is flat, there are no smaller peaks besides the dominant ones?

In the figure 2, the chromatograms of chlorogenic acid standard (CGA) to λmax = 320 nm and of caffeine (CAF) to λmax = 280 nm shown only one compound; to these long waves, it was possible to see two or three compounds around of them, mainly for the green coffee infusion.

  1. I don't see that there is any diagram or table showing the results and the dependence of the content of caffeine and chlorogenic acid depending on % roasted.

The contents of chlorogenic acid, caffeine and melanoidins in the green beans of C. arabica varieties and the coffee bean mixture, and medium roasted coffee (MRC) and  dark rosted coffee (DRC) are included in the table 3.

In my opinion, the current form of the work does not satisfy the criteria to be published. I suggest that the paper goes through a major review and then return to the reviewer for reading.

The manuscript molecules-2422094 entitled: “Chemical and biological characterization of green and processed coffee beans from Coffea arabica varieties” deals with the examination of coffee quality (physical, chemical, and biological characteristics).

I think the work is well conceived, interesting, but not well presented. I suggest that the authors edit the text, paying particular attention to the results and discussion, so that one can understand the significance of the obtained results. English must be improved!

  1. Authors must immerse themselves in their work and present the results in a meaningful way. It is very confusingly written. Many sentences are too long. In many places, it is not clear why the data from the literature are cited, they are not connected with the obtained results.

All the manuscript was reviewed by the authors,

  1. A comment related to Figure 2: is it possible to have only one compound at 330 nm, and one at 280, and that the rest of the baseline is flat, there are no smaller peaks besides the dominant ones?

In the figure 2, the chromatograms of chlorogenic acid standard (CGA) to λmax = 320 nm and of caffeine (CAF) to λmax = 280 nm shown only one compound; to these long waves, it was possible to see two or three compounds around of them, mainly for the green coffee infusion.

  1. I don't see that there is any diagram or table showing the results and the dependence of the content of caffeine and chlorogenic acid depending on % roasted.

The contents of chlorogenic acid, caffeine and melanoidins in the green beans of C. arabica varieties and the coffee bean mixture, and medium roasted coffee (MRC) and  dark rosted coffee (DRC) are included in the table 3.

Reviewer 2 Report

please add additional keyword - C. arabica beans

introduction section and aim of study - aim should has more details (specific research and subject of research)

introduction section - please write more information about the importance of coffee properties for the Mexican market and other recipients, some additional market analysis should be cited

materials and methods - please add information about source of coffee beans - private cultivation or specially established for research purposes?

 was a laboratory or industrial roaster used? please explain

how were the beans cooled? were grain parameters controlled during its cooling? please add more information

3.3.1 Infusion preparation - please add information about room temperature

results and discussion - tabels - please check journal requirements

conclusions -  please extend the summary in terms of the impact of the research on the importance of drinking coffee

supplementary materials should be use in main paper

references - please check journals abbrevations (an example European Food Research and Technology)

references - please check journal requirements (an example bold and ittalic style)

the paper should be check by native editor

Author Response

  1. Please add additional keyword - arabica beans

  1. arabica beans keyword was added

  1. Introduction section and aim of study - aim should has more details (specific research and subject of research).

The introduction and the aim of the study were improved.

  1. Introduction section - please write more information about the importance of coffee properties for the Mexican market and other recipients, some additional market analysis should be cited

Information about biological properties Mexican coffee market was added.

  1. Materials and methods - please add information about source of coffee beans - private cultivation or specially established for research purposes?

The cherries of Coffea arabica varieties, Typica, Bourbon, and Oro Azteca, employed in the present study, were harvested in the 2020-2021 and 2021-2022 cycles in the plantations of the Cooperative Cafeticultores Mephaa region of “La Montaña”, ….

  1. Was a laboratory or industrial roaster used? please explain

The mixture of coffee beans was processed in a 100MEX® brand industrial roaster with steel rotating cylinder.

  1. How were the beans cooled? were grain parameters controlled during its cooling? please add more information

The roasted beans are transferred onto plates and allowed to cool at room temperature.

  1. 3.1 Infusion preparation - please add information about room temperature

Infusions of each coffee were prepared in the laboratory at room temperature by triplicates, with 13.2 g of ground coffee beans in 200 mL of boiling water to 98 °C in a French press for 5 min.

  1. Results and discussion - tables - please check journal requirements

Tables were modified according to journal requirements

  1. Conclusions - please extend the summary in terms of the impact of the research on the importance of drinking coffee

  1. Supplementary materials should be use in main paper

Table with the spectroscopic data of chlorogenic acid was included in the paper

  1. References - please check journals abbreviations (an example European Food Research and Technology).

Journal abbreviations in the references were revised: Eur. Food Res. Technol, Food Res. Int., Food Chem. Toxicol., Nutr. Res.,  Braz. J. Plant Physiol., Eur. Food Res. Technol., Rev. Mex. de Cienc. Agric.

  1. References - please check journal requirements (an example bold and italic style)

Journal requirements from italic and bold letters, and abbreviations were reviewed

  1. Please add additional keyword - arabica beans

  1. arabica beans keyword was added

  1. Introduction section and aim of study - aim should has more details (specific research and subject of research).

The introduction and the aim of the study were improved.

  1. Introduction section - please write more information about the importance of coffee properties for the Mexican market and other recipients, some additional market analysis should be cited

Information about biological properties Mexican coffee market was added.

  1. Materials and methods - please add information about source of coffee beans - private cultivation or specially established for research purposes?

The cherries of Coffea arabica varieties, Typica, Bourbon, and Oro Azteca, employed in the present study, were harvested in the 2020-2021 and 2021-2022 cycles in the plantations of the Cooperative Cafeticultores Mephaa region of “La Montaña”, ….

  1. Was a laboratory or industrial roaster used? please explain

The mixture of coffee beans was processed in a 100MEX® brand industrial roaster with steel rotating cylinder.

  1. How were the beans cooled? were grain parameters controlled during its cooling? please add more information

The roasted beans are transferred onto plates and allowed to cool at room temperature.

  1. 3.1 Infusion preparation - please add information about room temperature

Infusions of each coffee were prepared in the laboratory at room temperature by triplicates, with 13.2 g of ground coffee beans in 200 mL of boiling water to 98 °C in a French press for 5 min.

  1. Results and discussion - tables - please check journal requirements

Tables were modified according to journal requirements

  1. Conclusions - please extend the summary in terms of the impact of the research on the importance of drinking coffee

  1. Supplementary materials should be use in main paper

Table with the spectroscopic data of chlorogenic acid was included in the paper

  1. References - please check journals abbreviations (an example European Food Research and Technology).

Journal abbreviations in the references were revised: Eur. Food Res. Technol, Food Res. Int., Food Chem. Toxicol., Nutr. Res.,  Braz. J. Plant Physiol., Eur. Food Res. Technol., Rev. Mex. de Cienc. Agric.

  1. References - please check journal requirements (an example bold and italic style)

Journal requirements from italic and bold letters, and abbreviations were reviewed

  1. Please add additional keyword - arabica beans

  1. arabica beans keyword was added

  1. Introduction section and aim of study - aim should has more details (specific research and subject of research).

The introduction and the aim of the study were improved.

  1. Introduction section - please write more information about the importance of coffee properties for the Mexican market and other recipients, some additional market analysis should be cited

Information about biological properties Mexican coffee market was added.

  1. Materials and methods - please add information about source of coffee beans - private cultivation or specially established for research purposes?

The cherries of Coffea arabica varieties, Typica, Bourbon, and Oro Azteca, employed in the present study, were harvested in the 2020-2021 and 2021-2022 cycles in the plantations of the Cooperative Cafeticultores Mephaa region of “La Montaña”, ….

  1. Was a laboratory or industrial roaster used? please explain

The mixture of coffee beans was processed in a 100MEX® brand industrial roaster with steel rotating cylinder.

  1. How were the beans cooled? were grain parameters controlled during its cooling? please add more information

The roasted beans are transferred onto plates and allowed to cool at room temperature.

  1. 3.1 Infusion preparation - please add information about room temperature

Infusions of each coffee were prepared in the laboratory at room temperature by triplicates, with 13.2 g of ground coffee beans in 200 mL of boiling water to 98 °C in a French press for 5 min.

  1. Results and discussion - tables - please check journal requirements

Tables were modified according to journal requirements

  1. Conclusions - please extend the summary in terms of the impact of the research on the importance of drinking coffee

  1. Supplementary materials should be use in main paper

Table with the spectroscopic data of chlorogenic acid was included in the paper

  1. References - please check journals abbreviations (an example European Food Research and Technology).

Journal abbreviations in the references were revised: Eur. Food Res. Technol, Food Res. Int., Food Chem. Toxicol., Nutr. Res.,  Braz. J. Plant Physiol., Eur. Food Res. Technol., Rev. Mex. de Cienc. Agric.

  1. References - please check journal requirements (an example bold and italic style)

Journal requirements from italic and bold letters, and abbreviations were reviewed

  1. Please add additional keyword - arabica beans

  1. arabica beans keyword was added

  1. Introduction section and aim of study - aim should has more details (specific research and subject of research).

The introduction and the aim of the study were improved.

  1. Introduction section - please write more information about the importance of coffee properties for the Mexican market and other recipients, some additional market analysis should be cited

Information about biological properties Mexican coffee market was added.

  1. Materials and methods - please add information about source of coffee beans - private cultivation or specially established for research purposes?

The cherries of Coffea arabica varieties, Typica, Bourbon, and Oro Azteca, employed in the present study, were harvested in the 2020-2021 and 2021-2022 cycles in the plantations of the Cooperative Cafeticultores Mephaa region of “La Montaña”, ….

  1. Was a laboratory or industrial roaster used? please explain

The mixture of coffee beans was processed in a 100MEX® brand industrial roaster with steel rotating cylinder.

  1. How were the beans cooled? were grain parameters controlled during its cooling? please add more information

The roasted beans are transferred onto plates and allowed to cool at room temperature.

  1. 3.1 Infusion preparation - please add information about room temperature

Infusions of each coffee were prepared in the laboratory at room temperature by triplicates, with 13.2 g of ground coffee beans in 200 mL of boiling water to 98 °C in a French press for 5 min.

  1. Results and discussion - tables - please check journal requirements

Tables were modified according to journal requirements

  1. Conclusions - please extend the summary in terms of the impact of the research on the importance of drinking coffee

  1. Supplementary materials should be use in main paper

Table with the spectroscopic data of chlorogenic acid was included in the paper

  1. References - please check journals abbreviations (an example European Food Research and Technology).

Journal abbreviations in the references were revised: Eur. Food Res. Technol, Food Res. Int., Food Chem. Toxicol., Nutr. Res.,  Braz. J. Plant Physiol., Eur. Food Res. Technol., Rev. Mex. de Cienc. Agric.

  1. References - please check journal requirements (an example bold and italic style)

Journal requirements from italic and bold letters, and abbreviations were reviewed

Round 2

Reviewer 1 Report

The manuscript is now much clearer to read and can be accepted in this form.